# It depends on how you tell: a qualitative diagnostic analysis of the implementation climate for community-wide mass drug administration for soil-transmitted helminth

Euripide Avokpaho,[1] Sarah Lawrence,[2] Amy Roll,[3,4] Angelin Titus,[5] Yesudoss Jacob,[5] Saravanakumar Puthupalayam Kaliappan,[5] Marie Claire Gwayi-Chore,[3,4] Félicien Chabi,[1] Comlanvi Innocent Togbevi,[1] Abiguel Belou Elijan,[1] Providence Nindi,[6] Judd L Walson,[3,4] Sitara Swarna Rao Ajjampur,[5] Moudachirou Ibikounle,[1,7] Khumbo Kalua,[6] Kumudha Aruldas,[5] Arianna Rubin Means [ID] [3,4]

EA and SL are joint first authors.

For numbered affiliations see end of article.

**Correspondence to**
Dr Arianna Rubin Means;
aerubin@uw.edu

## ABSTRACT

**Objectives** Current soil-transmitted helminth (STH) morbidity control guidelines primarily target deworming of preschool and school-age children. Emerging evidence suggests that community-wide mass drug administration (cMDA) may interrupt STH transmission. However, the success of such programmes depends on achieving high treatment coverage and uptake. This formative analysis was conducted to evaluate the implementation climate for cMDA and to determine barriers and facilitators to launch.

**Settings** Prior to the launch of a cMDA trial in Benin, India and Malawi.

**Participants** Community members (adult women and men, children, and local leaders), community drug distributors (CDDs) and health facility workers.

**Design** We conducted 48 focus group discussions (FGDs) with community members, 13 FGDs with CDDs and 5 FGDs with health facility workers in twelve randomly selected clusters across the three study countries. We used the Consolidated Framework for Implementation Research to guide the design of the interview guide and thematic analysis.

**Results** Across all three sites, aspects of the implementation climate that were facilitators to cMDA launch included: high community member demand for cMDA, integration of cMDA into existing vaccination campaigns and/or health services, and engagement with familiar health workers. Barriers to launching cMDA included mistrust towards medical interventions, fear of side effects and limited perceived need for interrupting STH transmission. We include specific recommendations from community members regarding cMDA distribution sites, personnel requirements, delivery timing and incentives, leaders to engage and methods for mobilising participants.

**Conclusions** Prior to launching the cMDA programme as an alternative to school-based MDA, cMDA was found to be generally acceptable across diverse geographical and demographic settings. Community members, CDDs

## STRENGTHS AND LIMITATIONS OF THIS STUDY

⇒ This study conducted focus group discussions (FGDs) across three diverse settings, creating the opportunity to understand similarities and differences in the implementation climate for community-wide mass drug administration and soil-transmitted helminth (STH) transmission interruption.

⇒ Some participants may have heard about the intervention before participating in FGDs, which may pose threats to social desirability and response biases.

⇒ Although a large number of FGDs were conducted across heterogeneous settings, the generalisability of study findings may not be translatable to other STH-endemic areas.

and health workers felt that engaging communities and tailoring programmes to the local context are critical for success. Potential barriers may be mitigated by identifying local concerns and addressing them via targeted community sensitisation prior to implementation.

**Trial registration number** NCT03014167; Pre-results.

## INTRODUCTION

Neglected tropical diseases (NTDs) affect some of the world's most impoverished populations and contribute to a variety of morbidities that exacerbate existing health and economic inequities.[1] Infections with one group of NTD, soil-transmitted helminths (STH), are associated with anaemia, cognitive development delay, growth stunting, preterm birth and other adverse outcomes.[2] WHO guidelines recommend control of STH morbidities via annual or biannual

deworming of preschool and school-age children and other high-risk groups, including pregnant women and adolescent girls.[3] However, in many settings, the current STH strategy would likely need to be continued until significant economic development and universal water, sanitation and hygiene (WASH) access are broadly available to stop transmission of STH.[4] Emerging evidence suggests that it may be possible to interrupt transmission of STH by expanding deworming to treat individuals of all ages via community-wide mass drug administration (cMDA).[5 6]

The current standard of care for STH is school-based MDA to reach preschool and school-age children, and many school-based deworming programmes have been successfully implemented for decades. Transitioning from school-based MDA to cMDA for community-level STH transmission interruption will require adapting long withstanding programmes or designing new platforms for reaching much larger target populations. These transitions should be approached carefully, as they will likely affect community member and health worker attitudes towards and engagement in new programmes. The success of cMDA in interrupting transmission of STH is indeed predicated on programmes attaining high treatment coverage (drug receipt) and uptake (drug ingestion).[7 8] Many STH-endemic communities have a long history of participating in other community-based mass distribution programmes, including campaigns for lymphatic filariasis (LF), onchocerciasis, trachoma, malaria bed net distribution and/or mass immunisation programmes such as polio campaigns.[9] Factors that influence participation in mass campaigns include recipient trust in programmes and efforts to tailor programmes to local conditions.[10] Negative campaign experiences or perceptions can compromise the success of future programmes, particularly those requiring high coverage over multiple years to reach targeted transmission endpoints.[11 12]

Formative qualitative research can be used to understand community-member and implementer perceptions of past, ongoing or prospective community-based campaigns. Diagnostic analyses, an application of formative evaluations, are particularly helpful in illuminating processes that can facilitate or impede implementation. Diagnostic analyses help to identify determinants of current practices, potential barriers and facilitators to implementing new interventions, and the perceived feasibility or utility of a new implementation strategy. This formative evidence can help researchers and implementers understand potential implementation challenges and, ideally, address them prior to intervention launch.[13] In this study, we perform a diagnostic analysis of the implementation climate to proactively identify factors influencing the launch of cMDA for STH transmission interruption, including[1] perceptions of current deworming practice,[2] potential barriers and facilitators to transitioning from school-based MDA to cMDA delivery, and[3] perceived effectiveness and need for cMDA.[14]

## METHODS

This analysis was conducted at the outset of the DeWorm3 Project, a large hybrid type I community cluster randomised trial in Benin, India and Malawi (table 1). Launched in 2017, the currently underway DeWorm3 Project aims to determine the feasibility of interrupting STH transmission using twice annual cMDA treating eligible individuals of all ages, relative to standard-of-care school-based MDA. More information about the DeWorm3 cluster randomised trial design has been described in detail elsewhere.[15–17]

### Sampling strategy

Key stakeholders shaping the implementation climate for cMDA include community members and local health

| Table 1 | Overview of study sites | | |
|---|---|---|---|
| | **Benin** | **India** | **Malawi** |
| Site | Commune of Comè | Vellore and Thiruvannamalai Districts, Tamil Nadu | Mangochi District |
| Geographic area of site (km²) | 148 | 477 | 289 |
| Total no of households | 24 378 | 36 536 | 27 750 |
| Population size | 94 969 | 140 932 | 121 819 |
| Standard of care | Annual school-based MDA targeting children 5–14 years of age | Biannual school-based MDA on National Deworming Days targeting children 1–19 years of age | Annual school-based MDA and Child Health Days targeting children 1–14 years of age |
| cMDA workforce | Community drug distributors (CDDs), coordinated by the Ministry of Health | CDDs and Accredited Social Health Activists, women working as health educators and promoters in their communities | Community health workers (Health Surveillance Assistants) who also fill the rolls of CDDs, coordinating with teachers |

cMDA, community-wide mass drug administration.

**Table 2** Sampling strategy by stakeholder group

| Stakeholder | Targeted sample size (per FGD) | Sampling strategy |
|---|---|---|
| Community members | | |
| Adult women (15+ years of age) | 5–10 | Purposive sampling (India) Random sampling (Benin* and Malawi) |
| Adult men (15+ years of age) | 5–10 | |
| Local leaders | 5–10 | |
| Children (12–15 years of age) | 5–10 | |
| Health centre staff and CDD supervisors | 5–10 | Purposive quota sampling |
| CDDs | 10–15 | Purposive quota sampling |

*Purposive quota sampling was used to sample local leaders in Benin and India.
CDDs, community drug distributors; FGD, focus group discussion.

workers. Focus group discussions (FGDs) were conducted separately with groups of community members, including adult women and men (over 15 years), community leaders, and children (ages 12–15 years of age), local health workers, including community drug distributors (CDDs) and Ministry of Health (MOH) health facility workers who often serve as CDD supervisors.

Prior to trial randomisation (eg, before designations of intervention or control clusters were made), four clusters were randomly selected in each site to conduct community-level FGDs. In each cluster, one FGD was conducted within each community member strata (four total), two FGDs among drug distributors and one FGD among local MOH health facility workers. The sampling strategy for identifying and recruiting community members for FGDs within each cluster differed slightly by site (table 2). In India, purposive sampling was employed, in which village leaders/influencers identified potential participants. In Malawi, community members were selected to participate via pseudorandomisation from a pool of individuals who attended outreach meetings at the chiefs/headmen's residence. The first five randomly approached individuals from each demographic strata who agreed to participate were invited to attend FGDs within the next week (except children, for whom parents/caregivers were approached). In Benin, community members were selected from a randomly generated list of potential participants from a baseline census database. The research team contacted the household heads by telephone and invited a specific individual (woman, man or child) to participate in an FGD. No more than one individual per household was selected to participate in an FGD in a given cluster. Transportation was offered to individuals who needed access to the FGD location. In Benin and India, local leaders were chosen using purposive quota sampling, during which DeWorm3 study teams invited key leaders in each selected cluster. Leaders differ setting by setting, wherein in some countries key leaders primarily include village chiefs while in other areas key leaders are primarily religious leaders. Purposive quota sampling was also used to invite CDDs and health workers from local health facilities located in each cluster to participate in FGDs.

## Data collection

This diagnostic analysis study design is informed by the Consolidated Framework for Implementation Research (CFIR), a meta-theoretical framework of 38 constructs that provides a typology of constructs for characterising potential determinants (barriers and facilitators) to implementation from the perspective of individuals involved in implementation.[18] The CFIR has been used widely in low-and-middle-income countries to identify factors that could influence or are actively influencing successful implementation.[19] CFIR constructs are organised according to five major domains influencing implementation and implementation effectiveness including[1] the intervention,[2] the inner setting,[3] the outer settings,[4] the individuals involved and[5] the process for accomplishing the intervention. While the CFIR can be used at any stage of implementation, when applied preimplementation, the CFIR can help proactively identify opportunities and challenges facing implementation and inform adaptations to implementation strategies for the local context.[18 20]

We drew on the CFIR to inform the design of four semistructured interview guides with a mix of respondent and informant style questions, tailored to each stakeholder group (one question guide was used for all adult community members). In this study, we identified a priori 23 CFIR constructs across all five domains that we hypothesised would influence the implementation climate for cMDA and which were appropriate for use during formative diagnostic research prior to implementation (online supplemental appendix 1).[20] The question guides were piloted and adapted slightly by changing word choice or sentence construction as necessary within each site to ensure that the questions were clear, meaningful and culturally appropriate. Site adapted question guides were thereafter translated into local languages including Yao (Malawi) and Tamil (India), and the official language (French) in Benin. FGD facilitators in Benin adapted the French question guide to local languages, including Watchi and Pédah when necessary, during FGD facilitation.

All participants provided written consent prior to the start of the FGD. The parents or caregivers of participating

children similarly provided written consent and children ages 12–15 also provided written assent. Consent and assent could also be provided by a thumbprint in the presence of a witness. FGDs were conducted in private locations with both a facilitator and notetaker present and all FGDs were audiorecorded with participant permission.

## Analysis

Audio files were transcribed verbatim in the local or official language. For each transcript, two 1 min random spot checks were conducted on each audio file for quality assurance. All transcripts were then translated into English. All transcriptions and their translations were reviewed by a second individual fluent in both English and the local language for quality assurance. Transcripts were imported into ATLAS.ti V.8 (Scientific Software Development, Berlin, Germany), which was used to manage data analysis. Coders were based in each DeWorm3 site as well as at the central level (University of Washington, Seattle). For data collected in Benin and India, two primary coders were assigned to each transcript, with a third coder designated as the 'tie-breaker'. When possible, at least one coder was based at the site in which the data were collected, and the other coder was a member of the DeWorm3 central team. For data collected in Malawi, a single primary coder from the central level coded the data while a secondary coder at the site reviewed and validated the findings, due to coder availability. Each primary coder independently read and coded each transcript primarily using a deductive approach and a CFIR-based codebook. Coding teams from each country and the central level met via conference calls to iteratively refine code definitions and code inclusion/exclusion criteria until a final codebook was established. After a transcript was coded, the coders assigned to the transcript met via conference call for consensus meetings to discuss where applied codes diverged. When necessary, a third coder weighed in where consensus between primary coders was not reached. Data saturation was reached as no new themes emerged during iterative review of the collected data.

The final coded transcripts were used to create case memos that were grouped by stakeholder category and site. The case memos included a summary of how the code was applied for a given stakeholder group, a justification for the summary provided noting code patterns and latent messages, and specific quotes highlighting how the code was applied. The summaries, patterns and themes from the coded transcripts and case memos were used to guide thematic analysis, an analytical method that is useful for summarising key features of large datasets using a clearly structured approach.[21 22]

## Patient and public involvement

Community members and health workers living in the sampled STH endemic areas were not involved in design, conduct or reporting of this qualitative study. However, all feedback from community members was used to shape

a subsequent community-based intervention within a larger clinical trial.

## RESULTS

In this study, 48 FGDs were conducted with community members—4 FGDs (one per cluster) for each stakeholder group: adult women, adult men, children and local leaders, totalling 16 in each site, 13 with CDDs (2 each in Benin and Malawi, 9 in India), and 5 with CDD supervisors (2 in Benin and 3 in India).

Across FGDs and settings, key themes emerged within four CFIR domains and are presented accordingly below: intervention characteristics, inner settings, characteristics of individuals and process. Factors positively influencing the implementation climate for cMDA across sites included community member demand for community-wide (vs school-based) MDA, integration of MDA into existing vaccination campaigns and/or health services, and engagement with health workers (including trained CDDs) rather than community volunteers. Factors negatively affecting the implementation climate across sites included mistrust and resistance toward medical interventions, fear of side effects and limited perceived need. Additional process recommendations emerged as key themes that varied slightly across sites and included suggestions regarding MDA distribution sites and distributors, treatment costs/financial incentives, engaging leaders, and engaging participants through sensitisation and mobilisation efforts.

### Intervention characteristics
#### Relative advantage: cMDA is preferable to school-based MDA
The CFIR construct of relative advantage captures participant perceptions regarding the benefits of implementing one intervention compared with an alternative.[18] Across community member and health worker/CDD groups and sites, participants identified a preference for cMDA as compared with school-based deworming programmes for several reasons. Participants stressed that providing STH treatment to both children and adults is the only way to prevent STH reinfection.

Across stakeholder groups, participants also highlighted that children who were not enrolled in school would be able to receive treatment through cMDA. Adult community members in Benin and Malawi were particularly concerned that school-based MDA campaigns do not always provide parents with treatment information prior to distribution and often administer the medications without parental consent or trust. Additionally, they thought uptake would be improved if parents are involved in treatment administration.

It's better to go through the parents to reach the kids. Parents know how to approach their children, manage them and make them understand the benefit of the thing [medicine]. The child will easily take the

**Table 3** Recommendations to optimise the implementation climate for newly launched cMDA

| Recommendation category | Benin | India | Malawi |
|---|---|---|---|
| MDA distribution mode | Door-to-door distribution | Door-to-door distribution preferable; potential for 3–4 central distribution sites in some communities | Door-to-door distribution |
| Intervention cost/financial incentives for participation | Free, but need to address rumours about nefarious intentions behind free MDA distribution | Free treatment preferable to most participants; need to address fears of perceived poor-quality medications provided by government programmes. Financial incentives should not be given for MDA participation, but incentives such as combs and soap were suggested | Free, but communities with past exposure to research studies might expect financial incentives for MDA participation |
| Community drug distributor preferences | Health workers (health facility workers or CDDs) who are familiar to community members | Trained health workers (nurses, doctors, ASHAs) who are familiar. Individuals without training should not be distributors | Health workers (including HSAs) who are familiar to community members. Volunteers are less respected and should not be distributors |
| Duration and time of distribution | Distribution over multiple days to accommodate different household schedules and reach the greatest number of people. Rainy season and market days should be avoided. Must consider work schedules and implement flexible distribution times | Distribution over multiple days. Evening or early morning preferred distribution time to accommodate work schedules | Distribution over multiple days to accommodate different household schedules and reach the greatest no of people |
| Key leaders to engage prior to cMDA | Village chiefs, religious leaders, and health workers | President and ward councillor of community (Panchayat), other health workers (Anganwadi workers), and teachers | Village chiefs, local leaders, religious leaders, local NGOs, HSAs, and teachers |
| Community education topics to engage MDA participants | Educate community about purpose and potential side effects of treatment | Educate community about purpose, advantages, and potential side effects of treatment, and proper dosage for different people (eg, children, elders) | Educate community about purpose and potential side effects of treatment; sensitisation must be done more than 1 day in advance to allow decision-making time |
| Mechanisms for engaging community members | Utilise radio, phones, community meetings and word of mouth to share information. Ring gongs at distribution time | Utilise radio, loudspeaker announcements, flyers, health documentaries, TV news, community meetings (women's groups), and community dramas to share information. Beat drums at distribution time | Utilise radio, phones, loudspeaker announcements, dramas, community meetings, door-to-door outreach, to share information |

ASHA, Accredited Social Health Activists; CDDs, community drug distributors; c-MDA, community-wide mass drug administration; NGOs, non-governmental organisation; TV, television.

medicine without any effect. (Cluster 26, Women, Benin)

Across FGDs and sites, participants were enthusiastic that cMDA could interrupt STH transmission and increase parental engagement with the intervention, particularly to enable parental consent, and allow them to encourage and confirm their child's uptake.

Design quality describes stakeholder recommendations for how to bundle, present and assemble the intervention.[18] Across sites and stakeholder groups, campaigns that delivered services door-to-door were considered more desirable than those that used a central distribution site (table 3). In India, community participants reflected on past experiences with LF MDA campaigns that were door-to-door whereas participants in Benin reflected on experiences with door-to-door vaccination campaigns and bed net distributions at local health centres. Long waiting times, associated with lost income and productivity, were identified as primary barriers to central distribution sites. One female participant in Benin reported she would not wait around all day for someone to distribute MDA but instead would just purchase the medications herself,

given their low costs. Additionally, participants stressed that door-to-door campaigns improve equity by increasing the likelihood of reaching those unable to travel due to financial or physical barriers.

> The HSAs should go door by door to give people the medicine as some people, for example old and crippled, may not be able to go and access the drugs. But if they go door by door, then everyone receives the drugs and not only those who walk. (Cluster 21, Women, Malawi)

Community members and leaders across sites, participants preferred to receive treatment from individuals perceived to be health professionals, especially a familiar health worker or CDD, or someone working with a well-respected non-governmental organisation (NGO). Participants believed that increased health worker engagement could alleviate community mistrust linked with fear of adverse events by medicalising the distribution process and making community members feel safer, thereby increase treatment coverage. Men in Malawi stressed that health volunteers are often poorly respected and

mistrusted, while clinically trained health professionals are typically more respected. Of paramount importance for adult community members and CDDs was that distributors are known members of the community.

> When community members see new faces during a project, they tend to be resistant, so it is better to use people from the area and not strangers. If not, this may not be successful. (Cluster 21, Local leader, Malawi)

> But above all, it is necessary to involve health workers, who the population trusts….Many are afraid because they do not see us, they do not see the health workers on the ground. (Health Center Staff, Benin)

In Benin, local leaders noted that when NGOs engage in cMDA, it is important they are well-respected and have well-recognised logos that community members are familiar with and trust based on their prior work. Regardless if cMDA is administered by a health professional, volunteer, or NGO, community members across sites noted that their willingness to participate in cMDA is driven by their perception that they have been fully and accurately informed about cMDA, and that they have had time to ask questions.

> Even if it [deworming medication] is given for free, they will not eat it unless it has been explained and given. If they are told they will benefit…with awareness in the villages, they will eat it. (Cluster 12, CDD, India)

Community members expressed that they wanted to be treated with dignity and that their participation in community-wide public health campaigns of any kind should not be taken for granted.

> Whether they eat the tablet or not it depends on to what extent this information reaches the people. It depends on how you tell. (Cluster 34, Men, India)

> The messages about drugs should be given to us in good time and not just tell us like today that tomorrow we will have a drug administration activity. Many people need time to ask questions and clear their myths before they get treated. Some people tend to refuse medicine because of fear of side effects, so when you sensitize them for a long time, they tend to listen and at the end the program becomes successful. So avoid short notice messages, people are difficult. They need enough time to understand what is happening. (Cluster 21, Men, Malawi)

In Malawi, local leaders reported that community members want to be followed up with after distribution to monitor for adverse events or continued engagement with distribution programmes to foster trust in future campaigns. Without this, the leaders feared that negative rumours might proliferate, or communities might feel as though they only received treatment for research purposes, rather than for their well-being.

## Intervention complexity: cMDA is complex, but still feasible to implement

The CFIR construct of complexity is defined as the perceived difficulty of implementing an intervention.[18] Across groups, participants were concerned about the timing of cMDA, the distance to distribution sites if cMDA is centrally located as opposed to delivered door-to-door, and whether or not they would have sufficient notice about cMDA before the campaign begins. Many community members suggested optimal distribution times, which varied by site depending on common work schedules and holidays. Concerns regarding health worker/CDD knowledge and accommodation of community members' schedules were prevalent across FGDs but some participants stressed they would change their schedules to be present for distribution if informed by community leaders.

> Even if someone has a plan to go to the field or to the market, three days before the distribution of the drug, they will cancel their plan and come and listen to what the village chief invited them to do…if everyone is not informed, it [MDA] cannot succeed. (Cluster 10, Women, Benin)

Adults and children across sites recommended that distribution over multiple days within a community to reach the greatest number of people.

> The period of drug administration should be long so that everyone is able to receive treatment. Some people may not be home during the time that you have set to administer the drugs and as such if done for maybe only a day, it means those people will not receive the drugs. But if it is for some more days then everyone will be treated. (Cluster 21, Men, Malawi)

### Inner setting

## Implementation climate: initial mistrust of MDA is likely, but demand and perceived need will counter this

The CFIR construct implementation climate captures comments related to the community member's receptivity to implementation, and the extent to which implementation is supported.[18] The core component of implementation climate discussed across FGDs were factors that influence community member trust in treatment campaigns. Participants across the sites anticipated high levels of initial mistrust and potential resistance towards newly launched cMDA for STH. This initial mistrust is driven by personal and anecdotal evidence of adverse side effects such as fatigue, stomachaches and fever after previous school-based deworming MDA campaigns.

> Other pupils received the medical treatment before they ate a meal, hence they vomited. So those that did not receive the medical treatment were afraid of vomiting too if they took the medicine. (Cluster 19, Children, Malawi)

> They will eat [medication] based on the trust. They will eat [medication] based on your approach,

otherwise they may take and keep it aside some-where…. (Cluster 17, Men, India)

In Malawi, limited follow-up by transitory MDA programmes and research projects was also noted as fueling mistrust of community health programmes. Similarly, in one CDD FGD in India, participants identified mistrust of government programmes as a potential barrier to MDA campaigns where medications are provided for free. CDDs explained that community members perceive government provided medications to be of poorer quality and therefore less effective with greater risks of side effects; therefore, those who can afford to purchase their own medications from pharmacies will often do so.

For people who can, they will get it [deworming medicine] from the medical shop. Whatever is given through the government they will keep it aside and they will not use it. (Cluster 15, CDD, India)

In Benin, some participants from the men's, women's, and CDD FGDs shared concerns that drugs used in such campaigns might be given by Westerners with malintent.

The majority of the population does not understand. They think that the drugs are poisoned in order to reduce the African population. (Cluster 10, Men, Benin)

Similarly, in Malawi, CDDs identified rumours and misinformation as major barriers to delivering cMDA with high coverage. Specific rumours include that stool collected for STH surveillance would be used for Satanist practices, rather than medical purposes, and that school-based deworming programmes provide contraceptives to young children to reduce population growth.

While participants noted that mistrust and resistance might initially be high following a transition to cMDA for STH, there was still a strong perceived demand for deworming of all ages, and a sentiment that community sensitisation could overcome these concerns.

…we are looking forward to this [community-based MDA] and we would like this to be a regular treatment. People are suffering from intestinal worms and only children receive the treatment. So this project [DeWorm3] will help all of us to receive treatment. (Cluster 21, Men, Malawi)

## Compatibility: community-based MDA is highly compatible with existing health infrastructure

The CFIR construct compatibility is highly related to implementation climate, capturing the alignment between the innovation and existing values and priorities.[18] Health facility workers and CDDs across sites noted cMDA should be integrated into existing community programmes or, at a minimum, coordinate with ongoing community-based activities to improve treatment coverage and mitigate risks of conflict with ongoing local health programmes. CDDs in India and community leaders in Benin identified community-based vitamin A and iron distribution and childhood vaccination campaigns as ideal programmes to integrate with cMDA. In India, CDDs also suggested integrating cMDA with existing indoor residual spraying programmes for vector control.

The voluntary workers who go house to house to spray mosquitoes, we can make use of them to give the tablet…Earlier they were going once a month or once a week, but now they go daily house to house. We can give through them. (Cluster 17, CDD, India)

## Available resources: training, storage and hygiene infrastructure are key resources for MDA implementation

The CFIR construct available resources describes the financial and material resources available (and desired) for implementation, including training and education, space, time, and money.[18] Health facility workers and CDDs across sites stressed that existing resources may not be sufficient for delivery of cMDA. CDDs in particular were concerned about receiving adequate training and access to resources to take home for self-review. CDDs and health workers highlighted that they wanted more than a single 1-day training prior to MDA, in order to provide adequate time to practice and apply skills in a training environment. Across sites, CDDs noted the importance of training before distribution.

Other key resources identified by CDDs and health workers in Benin included medication storage in the field, community education materials, shelter during inclement weather, as well as food, water and financial incentives for CDDs. CDDs in Benin were particularly concerned with timely payment for their work. In Malawi, local leaders noted that in the past villagers have felt burdened by volunteering for health programme implementation without compensation. They also noted that villagers might expect payment for participating in MDA, given past experiences with research projects providing stipends. In India, health facility workers wanted to ensure they would have adequate staffing to assist during MDA.

Lastly, CDDs and health centre personnel across sites, women in Benin, and leaders in India stressed that hygiene infrastructure needs to be improved and that investing in WASH as part of a broader STH elimination programme might, as a result, increase treatment coverage of cMDA by demonstrating long-term investments in community well-being.

The rules of hygiene are very important, very, very important. Without it, we cannot right away start distributing the drugs and say that we want to completely eradicate the transmission of worms, impossible. (CDD, Benin)

## Characteristics of individuals

The knowledge and beliefs CFIR construct is defined as individuals' attitudes towards and value placed on implementation and their familiarity with related facts, truths

and principles.[18] While participants strongly believed cMDA could eliminate STH transmission, some reservations about MDA rooted in knowledge and perceptions about deworming medications remain. For example, adult men and children in Benin and local leaders in Malawi raised concerns about the effects of treating people who may not be infected with STH.

> When you get the drugs and you do not have the worms, the tablet can still damage your organs such as organs of digestion or breathing. (Cluster 1, Children, Benin)

In India and Malawi, participants in the men's FGDs thought individuals who feel healthy might perceive themselves to be at low risk of STH and thus choose not to participate in cMDA. Women in Malawi reported this occurred during prior cMDA campaigns while women in India also noted individuals with limited literacy might not understand the need for treatment and be reluctant to participate in MDA campaigns.

### Self-efficacy: community members and CDDs will be key mobilisers for cMDA

The CFIR construct self-efficacy captures comments that reflect an individual's beliefs in their own abilities to achieve implementation goals.[18] Women and community leaders in Benin identified themselves as important catalysts in influencing the acceptability of MDA by working together and influencing their social networks.

> The process is simple as we have just understood, we will also explain to our brothers and sisters who will not accept that treatment is a good thing. We will tell them they should not be discouraged adding that there is good in it. It's up to us to explain to them. (Cluster 1, Women, Benin)

Health facility workers and CDDs also identified themselves as important contributors to ensuring successful MDA implementation, given their existing relationships with communities.

> We worked with them and they know us on the ground to be able to do the job, so there are no issues for community health workers. (CDD, Benin)

### Process
### Engaging: local leaders and sensitisation activities are essential for ensuring high treatment coverage

The CFIR construct engaging is defined as approaches to attracting and involving individuals in implementation, such as through social marketing or education campaigns.[18] Adult participants across sites, including local leaders, identified specific leaders as key facilitators of effective implementation (table 2). These leaders should be notified in advance of interventions taking place within their communities to gain their support and promote the intervention prior to implementation. Advance sensitisation with information

about the distribution time as well as potential intervention benefits and risks were identified as critical pieces of information that influence the implementation climate and, in effect, effective delivery of cMDA. Preferred engagement methods varied slightly across sites, however community members suggested that a variety of information-sharing mechanisms be used in advance of MDA to improve community member knowledge and buy-in (table 3).

## DISCUSSION

This diagnostic analysis highlights opportunities and challenges of launching cMDA for STH that are shared across geographic areas as well as important differences between them. Our findings build on the existing literature and demonstrate strong acceptability of cMDA for STH interruption, particularly as an alternative to school-based distribution to provide more equitable access to deworming treatment. When considering a transition from school-based distribution to cMDA, participants highlighted opportunities to integrate cMDA into existing community health programmes, such as vaccination campaigns, and the importance of engaging clinically trained, trusted drug distributors to mitigate fears of adverse events and increase treatment coverage. Using local CDDs from the same area or are directly known to the recipients, is associated with high MDA coverage in other settings.[23 24] While familiarity was important to participants in this study, they also stressed comprehensive clinical training for CDDs as essential for fostering trust during cMDA. While participants identified potential benefits of launching cMDA, they also noted key barriers that might limit implementation success. Similar to findings from studies exploring MDA barriers postimplementation, the primary barriers identified across sites included mistrust towards free drug distribution (especially those provided by community volunteers perceived to have no clinical training), fear of side effects, and limited perceived need for treatment without symptoms.[12 25–27] This information is essential for adapting interventions to fit the specific context and concerns of communities prior to making a significant change to public health programmes, such as changing from school-based delivery to community-wide delivery of deworming medicines.

In this formative evaluation, we synthesised recommendations from community members to assist in intervention optimisation at a site level, including preferred treatment time and distribution methods. Site-specific preferences and recommendations for implementation varied to small degrees across settings, including preferred distribution times, campaign duration, location, distributor qualifications and procedures for engaging leaders and community participants. Other studies have found that proactively identifying specific times when individuals are generally available to receive treatment is an essential facilitator of effective campaign delivery,[28–30] and when not completed can increase frustration with

community-based volunteer distributors[25] and the MDA campaign itself.[31] Where trial timeline and funding allow, formative evaluations such as this may facilitate proactive identification of potential barriers and of implementation processes to optimise community acceptability and adapt intervention delivery as needed. When timelines or funding are limited, like during cMDA implemented by national NTD programmes in limited-resources settings, brief surveys prior to MDA and interim analyses and may be conducted to tailor MDA implementation.

Emerging themes from community member, local leader, CDD and health worker FGDs were highly consistent. However, CDDs and health workers noted unique facilitators and barriers affecting their work, including time for training, resources in the field and timely compensation, similar feedback from implementers in other settings.[26] CDDs, health workers and community leaders also uniquely stressed the importance of packaging WASH interventions with MDA for STH elimination. These sentiments reflect advanced knowledge of STH transmission but may also reflect doubt that transmission interruption programmes predicated on broadly delivered MDA will interrupt transmission without also improving hygiene infrastructure. In fact, most of the benefits attributed to cMDA among these cadres was driven by beliefs that MDA will be more acceptable when delivered in the community, as opposed to its potential for interrupt transmission. Although evidence regarding the effectiveness of WASH on STH transmission interruption is still weak, it is also important to monitor how implementer and community enthusiasm for cMDA changes over time, potentially driven by the absence of WASH interventions.[32]

Participants across all geographical areas noted that myths and rumours could pose serious challenges to cMDA delivery, adding further to the literature documenting this obstacle in other settings.[10 11 26 33] Children in Benin and Malawi noted specific concerns about side effects of deworming treatment circulating within schools prior to and post-MDA campaigns. While side effects for albendazole are typically quite mild, albendazole is often coadministered to children with praziquantel as preventative chemotherapy for schistosomiasis. Praziquantel can cause relatively more severe side effects, including diarrhoea and vomiting.[34] Thus, prior experiences with MDA campaigns including other treatments might influence future perceptions about albendazole specifically or MDA generally.[35] Evidence suggests that effective health workers can overcome these individual-level perceptions[31]; addressing myths and rumours will require targeted and proactive community sensitisation and CDD training that openly discuss local myths and rumours preintervention.

Mistrust towards public health campaigns and government-run health programmes was prevalent across settings. These concerns were driven by prior negative experiences with medical interventions and programmes in which limited information was provided in advance of treatment, parents were minimally engaged in school-based MDA, and there were perceived concerns about drug quality. Potential strategies for overcoming these barriers include engaging local leaders,[11 29 36] targeted education campaigns,[26 37 38] community mobilisation[36 39 40] and engagement of trusted, trained personnel to administer preventive treatment.

During the design of our initial question guide (online supplemental appendix 1), we drew from selected constructs across all CFIR domains. However, during data analysis no constructs within the outer setting emerged as major facilitators or barriers to launch of cMDA campaigns. The outer setting domain is composed of constructs representing external influences on implementation and, perhaps because these data were collected at the community-level, respondents were more focused on individuals involved in implementation and implementation processes.[18] Additionally, because we conducted a formative study we did not link identified implementation determinants to observed implementation outcomes, however subsequent data collection activities in DeWorm3—once the trial is underway and outcome data are collected—will afford these opportunities.[20]

This study had several limitations. Some participants may have heard about DeWorm3 before participating in FGDs, which may have contributed to social desirability or response biases. While a large number of FGDs were conducted across heterogeneous settings, it is also possible that the study findings may not be generalisable to other STH-endemic areas.

## CONCLUSION

This study supports that cMDA, particularly as an alternative to school-based MDA, is generally acceptable across heterogenous settings and builds on the existing literature exploring facilitators and barriers of launching and implementing cMDA. Community engagement including STH education, understanding preferred distribution times and methods, involvement of local leaders and familiar health workers or CDDs are critical for implementation success. Potential barriers, including mistrust of free drug distribution, fear of side effects and limited perceived need of treatment can be addressed through community sensitisation and engagement of local leaders and trusted health workers. These findings were used to shape implementation activities during the DeWorm3 trial, in order to ensure high acceptability of the intervention and high cMDA coverage from the onset of the trial. Formative research exploring attitudes and community-derived recommendations should be conducted when possible to improve community acceptability of new interventions.

**Author affiliations**
[1]Institut de Recherche Clinique du Bénin, Abomey-Calavi, Benin
[2]Department of Pediatrics, University of Washington, Seattle, Washington, USA
[3]Department of Global Health, University of Washington, Seattle, Washington, USA
[4]The DeWorm3 Project, Seattle, Washington, USA

[5]The Wellcome Trust Research Laboratory, Division of Gastrointestinal Sciences, Christian Medical College Vellore, Vellore, India
[6]Blantyre Institute for Community Outreach, Blantyre, Malawi
[7]Centre de Recherche pour la lutte contre les Maladies Infectieuses Tropicales (CReMIT/TIDRC), Université d'Abomey-Calavi, Cotonou, Littoral, Benin

**Contributors** ARM, JLW, MCG-C, SSRA, KK and MI conceived of the initial study design. EA, SPK, FC, CIT, PN, MI, KK, KA, AT and YJ were involved in data collection and data processing. AR managed the data. AR, ABE, SL and ARM contributed to data analysis. SL, EA and ARM led the writing of the paper. All authors read and approved the final version of the manuscript. ARM serves as the guarantor of this research.

**Funding** The DeWorm3 study was funded by the Bill & Melinda Gates Foundation (https://www.gatesfoundation.org/) and (grant number OPP1129535).

**Disclaimer** The funders had no role in study design, data collection and analysis, decision to publish, or preparation of the manuscript.

**Competing interests** None declared.

**Patient and public involvement** Patients and/or the public were not involved in the design, or conduct, or reporting, or dissemination plans of this research.

**Patient consent for publication** Not applicable.

**Ethics approval** This study has been reviewed and approved by the Institut de Recherche Clinique du Bénin (IRCB) through the National Ethics Committee for Health Research (002–2017/CNERS-MS) from the Ministry of Health in Benin, The London School of Hygiene and Tropical Medicine (12013), The College of Medicine Research Ethics Committee (P.04/17/2161) in Malawi, and Christian Medical College, Vellore, in India (10392). The study was also approved by The Human Subjects Division at the University of Washington (STUDY00000180).

**Provenance and peer review** Not commissioned; externally peer reviewed.

**Data availability statement** Data are available on reasonable request. Qualitative data contains information that is personal and may be individually identifiable.

**ORCID iD**
Arianna Rubin Means http://orcid.org/0000-0002-4087-7080

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
