## [Reviewer comments · BMJ Open]

ARTICLE DETAILS

TITLE (PROVISIONAL)	"It depends on how you tell": A qualitative diagnostic analysis of the implementation climate for community-wide mass drug administration for soil-transmitted helminths
AUTHORS	Avokpaho, Euripide; Lawrence, Sarah; Roll, Amy; Titus, Angelin; Jacob, Yesudoss; PUTHUPALAYAM KALIAPPAN, SARAVANAKUMAR; Gwayi-Chore, Marie Claire; Chabi, Félicien; Togbevi, Comlanvi; Elijan, Abiguel Belou; Nindi, Providence; Walson, Judd L; Ajjampur, Sitara; Ibikounle, Moudachirou; Kalua, Khumbo; Aruldas, Kumudha; Means, Arianna

VERSION 1 – REVIEW

REVIEWER	Catherine Gordon QIMR Berghofer Medical Research Institute
REVIEW RETURNED	01-Mar-2022

GENERAL COMMENTS	Really well written and comprehensive article, and an important topic. Community engagement has been highlighted as a key reason for success of control programs elsewhere and this report will be of use for other health workers, NGOs, governments, and researchers when looking at implementing MDA. Lessons here can also be extrapolated to development of other intervention types. Line 83: one group of NTD Line 306: inner setting?
---

REVIEWER	Ana Lourdes Sanchez Brock Univ, Health Sciences
REVIEW RETURNED	01-Mar-2022

GENERAL COMMENTS	Manuscript ID: bmjopen-2022-061682 Title: "It depends on how you tell": A diagnostic analysis of the implementation climate for community-wide mass drug administration for soil-transmitted helminths" Review date Feb 28-March 01, 2022 Description of manuscript This is a qualitative study conducted in advance the launch The DeWorm3 Project, which, according to the project's website "will test the feasibility of this approach to interrupting the transmission of STH using a series of cluster randomized trials in Benin, India and Malawi". The Deworm3 project was multi-year project conducted between 2015-2021. The protocol was published in 2018 by Ásbjörnsdóttir et
--

al. "Assessing the feasibility of interrupting the transmission of soil-transmitted helminths through mass drug administration: The DeWorm3 cluster randomized trial protocol." PLoS NTD 12.1 (2018): e0006166.

The website also lists > 30 publications stemming from the project

The present manuscript recounts a qualitative study to "evaluate the implementation climate for community mass-drug administration (MDA) and to determine barriers and facilitators to launch."

General comment

For clarity, the title of the manuscript (ms) should reflect the type of study that was conducted, i.e., a qualitative study as "diagnostic analysis" is not sufficiently explicit—at least not to qualitative researchers. On the same note, the authors should strive for consistency when referring to the type study throughout the ms as there are too many versions: "diagnostic analysis", "Formative analysis", "Formative qualitative research", "formative evaluation". To a non-qualitative researcher this will be confusing.

One other issue with wording is stating that community-based deworming was an "alternative" instead of a replacement for school-based deworming. While the Consolidated Framework for Implementation Research (CFIR) does utilize the term alternative, in this case, as per the DeWorm3 principal hypothesis, it appears that the long-term vision is to move from school-based to population-wide mass deworming.

Specific comments

For better understanding and frame of reference to the reader, the work being presented should be clearly positioned within the DeWorm3 project. The project is mentioned but no real details are provided such as objectives and timelines. This is particularly important since the work presented was done prior to randomization and most (if not all) of DeWorm3 findings have already been published.

The methodology requires much more detail as well. For a methodological framework, the authors were guided by the Consolidated Framework for Implementation Research (CFIR). It would be important to know why was CFIR chosen over other methodologies and its practical strengths and shortcomings. Maybe the authors could cite similar studies on STH or NTDs that have validated CFIR.

As well, it is unclear why only focus group discussions (FGD) were done while in-depth interviews were deemed not necessary. One interesting aspect surfacing from FGD was "distrust in westerners" which would have merited an in-depth interview. Also, how were focus group participants recruited and how was the size of the focus groups determined? If the sampling strategy for a site or group was different, this requires an explanation. Table 2 is a good start as a methodological summary but it is as it stands, incomplete. Some of this information is presented loosely on the body of text but it's difficult for this reader to establish methodological comparisons and thus infer potential biases. I also noticed that Drug distributors are not reported as stakeholders in Table 2. Were they included? In terms of the questions asked during the FGD, the authors mention that the questions guides were piloted and adapted slightly. Would the authors please elaborate on what adaptations were made? Were

	there any substantial changes? Or was the interview instrument near perfect for eliciting the information they intended to capture? Other methodological questions are: how did the authors know when they reached saturation? Was a coding template (or codebook) used? What was the analyzing software, if any? Results/Discussion It may be more appropriate to organize results by CFIR domains to produce actionable findings (e.g., see table 4 and 5 from Keith et al 2017). Finally, the findings of the study identified factors which could affect the implementation either positively or negatively. However, they are presented as general recommendations (Table 3) instead of actionable items that authors sought to implement. Since the goal of CFIR is to “produce actionable evaluation findings intended to improve implementation in a timely manner” (Keith et al 2017), the results and discussion ought to be centered around that. Otherwise, this works reads as a theoretical exercise with a series of unsurprising findings and general recommendations that could have been gleaned from the literature. The ms discussion ends with “...while CFIR constructs should ideally be coupled to specific targeted outcomes, the formative nature of this study precludes linkage of implementation determinants to outcomes”. Possibly this is the crux of my concern and perhaps authors could elaborate. After all, authors stated at the outset that the study was conducted to evaluate the implementation climate for cMDA and to determine barriers and facilitators to launch”.
--	---

REVIEWER	Nor Asiah Muhamad Institute for Public Health
REVIEW RETURNED	07-Mar-2022

GENERAL COMMENTS	Dear Authors, Congratulation on your write up. Here are my comments  1. The description of your methodology part is confusing, the conceptual flaws in my opinion need to be improved. 2. As you are mentioning the formative qualitative research can be used to understand community-member and implementer perceptions in community-based campaigns and diagnostic analyses, it remains unclear how these processes works and can facilitate the application of formative evaluations. Maybe you can explain further in the method 3. The research gaps are not clearly described. What makes the study contribution and innovation unclear: is the gap seen in the methodological design of merely cross-sectional research? Or, in the used concepts and model? Please mention the gaps 4. How is randomisation done? What method is used?
---

VERSION 1 – AUTHOR RESPONSE

REVIEWER 1: Dr. Catherine Gordon, QIMR Berghofer Medical Research Institute

Comments to the Author:

Really well written and comprehensive article, and an important topic. Community engagement has been highlighted as a key reason for success of control programs elsewhere and this report will be of

use for other health workers, NGOs, governments, and researchers when looking at implementing MDA. Lessons here can also be extrapolated to development of other intervention types.

Thank you for this encouraging feedback. We greatly appreciate the reviewer's thoughtful comments.

Line 83: one group of NTD

We have updated the text per this suggestion.

Line 306: inner setting?

The results are organized by CFIR domain, inner setting is one of the five domains. We have added this detail to the introduction of the results section with the following italicized text: "Across FGDs and settings, key themes emerged within four CFIR domains *and are presented accordingly below*: intervention characteristics, inner settings, characteristics of individuals, and process."

REVIEWER 2: Dr. Ana Lourdes Sanchez, Brock Univ

Comments to the Author:

This is a qualitative study conducted in advance the launch The DeWorm3 Project, which, according to the project's website "will test the feasibility of this approach to interrupting the transmission of STH using a series of cluster randomized trials in Benin, India and Malawi". The Deworm3 project was multi-year project conducted between 2015-2021. The protocol was published in 2018 by Ásbjörnsdóttir et al. "Assessing the feasibility of interrupting the transmission of soil-transmitted helminths through mass drug administration: The DeWorm3 cluster randomized trial protocol." PLoS NTD 12.1 (2018): e0006166. The website also lists > 30 publications stemming from the project. The present manuscript recounts a qualitative study to "evaluate the implementation climate for community mass-drug administration (MDA) and to determine barriers and facilitators to launch."

General comments:

1. For clarity, the title of the manuscript (ms) should reflect the type of study that was conducted, i.e., a qualitative study as "diagnostic analysis" is not sufficiently explicit—at least not to qualitative researchers. On the same note, the authors should strive for consistency when referring to the type study throughout the ms as there are too many versions: "diagnostic analysis", "Formative analysis", "Formative qualitative research", "formative evaluation". To a non-qualitative researcher this will be confusing.

Thank you for your feedback. We have added "qualitative" to the title of the paper. In the introduction of the paper, we clarify that diagnostic analysis is indeed a function of formative evaluations, "Diagnostic analyses, an application of formative evaluations, are particularly helpful in illuminating processes that can facilitate or impede implementation". We hope that this definition in addition to a few additions throughout the paper such as adding the following

italicized text to “formative *diagnostic* research” will help clarify this for any non-qualitative researchers.

2. One other issue with wording is stating that community-based deworming was an “alternative” instead of a replacement for school-based deworming. While the Consolidated Framework for Implementation Research (CFIR) does utilize the term alternative, in this case, as per the DeWorm3 principal hypothesis, it appears that the long-term vision is to move from school-based to population-wide mass deworming.

Thank you for bringing up this interesting idea. Should a change in WHO guidelines be issued regarding STH, countries may still choose to use school-based deworming, community-wide deworming, or perhaps even a mix of both. Thus, we believe that community-wide MDA may not necessarily “replace” school-based deworming, but will be a viable implementation strategy for governments to consider implementing at scale or in combination with school-based deworming.

Specific comments

1. For better understanding and frame of reference to the reader, the work being presented should be clearly positioned within the DeWorm3 project. The project is mentioned but no real details are provided such as objectives and timelines. This is particularly important since the work presented was done prior to randomization and most (if not all) of DeWorm3 findings have already been published.

The DeWorm3 study is still underway and data related to the primary outcomes are blinded. We expect that the primary outcome analysis will take place in mid to late 2023. However, it is a large study with many secondary analyses and many publications have been generated thus far. Because this formative study took place prior to the launch of the DeWorm3 study, we do not believe extensive trial design details are necessary in this formative qualitative evaluation, however we have added the following italicized text to ensure that the reader has sufficient context. We also note that Table 1 provides extensive information about the site areas, demographic groups, DeWorm3 intervention, and standard of care.

“Launched in 2017, the currently underway DeWorm3 Project aims to determine the feasibility of interrupting STH transmission using twice annual cMDA treating eligible individuals of all ages, relative to standard-of-care school-based MDA. More information about the DeWorm3 cluster randomized trial design has been described in detail elsewhere (14-16).”

2. The methodology requires much more detail as well. For a methodological framework, the authors were guided by the Consolidated Framework for Implementation Research (CFIR). It would be important to know why was CFIR chosen over other methodologies and its practical strengths and shortcomings. Maybe the authors could cite similar studies on STH or NTDs that have validated CFIR.

As described in the methods section, we utilize the CFIR because it is a broadly used meta-theoretical determinants framework, well suited to identify barriers/facilitators to implementation and built upon a large number of existing theories and frameworks. We have added a citation to a systematic review from our team that explores use of the CFIR in low-and-middle-income countries to highlight its applicability in Benin, India, and Malawi where our research takes place.

3. As well, it is unclear why only focus group discussions (FGD) were done while in-depth interviews were deemed not necessary. One interesting aspect surfacing from FGD was “distrust in westerners” which would have merited an in-depth interview.

We appreciate this idea from the reviewer. We conducted extensive individual interviews with stakeholders above the level of the community (district, regional, national, and partner levels). These findings are reported elsewhere. Because of the large size of the DeWorm3 study (Table 1), it was not feasible to conduct individual interviews with community members and health workers. 48 FGDs is an extremely large sample size for a qualitative study, and thus individual interviews were not possible to implement in addition.

Also, how were focus group participants recruited and how was the size of the focus groups determined? If the sampling strategy for a site or group was different, this requires an explanation. Table 2 is a good start as a methodological summary but it is as it stands, incomplete. Some of this information is presented loosely on the body of text but it's difficult for this reader to establish methodological comparisons and thus infer potential biases.

Best practices in qualitative data collection indicate that focus groups should not be so large that is it challenging to have an open or vulnerable conversation amongst participants, but large enough that there can be discourse primarily between participants (as opposed to the facilitator playing a primary role). Thus, 5-15 participants were invited to each FGD, with fewer participants in community member FGDs being preferable due to the potential for private or vulnerable information to be disclosed.

We believe the methods section has adequate information on the differences in sampling strategy used for community members. We have added the following italicized text to provide additional detail about sampling of leaders, “The sampling strategy for identifying and recruiting community members for FGDs within each cluster differed slightly by site (Table 2). In India, purposive sampling was employed, in which village leaders/influencers identified potential participants. In Benin and Malawi, community members were randomly selected to participate from a pool of individuals who attended outreach meetings at the chiefs/headmen’s residence. The first five randomly approached individuals from each demographic strata who agreed to participate were invited to attend FGDs within the next week (except children, for whom parents/caregivers were approached). No more than one individual per household was selected to participate in an FGD in a given cluster. Transportation was offered to individuals who needed access to the FGD location. In Benin and India, local leaders were chosen using purposive quota sampling, *during which DeWorm3 study teams invited key leaders in each selected cluster to participate. Leaders differ setting by setting, wherein in some countries key leaders primarily include village chiefs while in other areas key leaders are primarily religious leaders.* Purposive quota sampling was also used to invite CDDs and health workers from local health facilities *located* in each cluster to participate in FGDs.”

I also noticed that Drug distributors are not reported as stakeholders in Table 2. Were they included?

Thank you for this comment. They were previously only abbreviated in Table 2. We have added the complete definition of “Community drug distributors” in the last row of Table 2.

In terms of the questions asked during the FGD, the authors mention that the questions guides were

piloted and adapted slightly. Would the authors please elaborate on what adaptations were made? Were there any substantial changes? Or was the interview instrument near perfect for eliciting the information they intended to capture?

Adaptations included simplifying and contextualizing the language in the guides to ensure that they were easily understood in the very different cultural and linguistic settings included in this study. As we note in the methods section, the purpose of the adaptations was to ensure that the questions “were clear, meaningful, and culturally appropriate.” As we note the guides were “slightly” adapted, mostly in regard to word choice or sentence construction, and there were no major adaptations or substantial changes. We have added this detail to the methods section. We would hesitate to consider any question guide “near perfect” and, like any qualitative study, during data analysis we learned important lessons about which questions elicited the most information and which questions were most challenging for participants to respond to. We are fortunate that we collected qualitative data repeatedly during the study, and findings from these baseline data were used to iteratively improve upon the question guide.

4. Other methodological questions are: how did the authors know when they reached saturation? Was a coding template (or codebook) used? What was the analyzing software, if any? **Thank you for these questions. We have added text stating, “*Data saturation was reached when no new themes emerged during iterative review of the collected data.*” In the analysis paragraph of the methods section, we describe our deductive approach to applying a codebook, which was CFIR-based and iteratively refined by coding teams from each country and the central level until a final codebook was established. We have added text noting coding was completed in ATLAS.ti version 8.**

Results/Discussion

5. It may be more appropriate to organize results by CFIR domains to produce actionable findings (e.g., see table 4 and 5 from Keith et al 2017).

Thank you for this suggestion. Our results are indeed already organized by CFIR domain, with the domains as sub-header titles.

6. Finally, the findings of the study identified factors which could affect the implementation either positively or negatively. However, they are presented as general recommendations (Table 3) instead of actionable items that authors sought to implement. Since the goal of CFIR is to “produce actionable evaluation findings intended to improve implementation in a timely manner” (Keith et al 2017), the results and discussion ought to be centered around that.

We appreciate the reviewer’s feedback and references to the Keith (2017) article, which greatly influenced our study design and team’s use of the CFIR. Table 3 is not a summary of key themes, but rather amplification of recommendations that stakeholders made for how to optimize the implementation of newly launched cMDA. To make it clear that these findings shaped study implementation, we have added the following italicized text to the conclusion section of the paper: “*These findings were used to shape implementation activities during the DeWorm3 trial, in order to ensure high acceptability of the intervention and high cMDA coverage from the onset of the trial.*”

Otherwise, this works reads as a theoretical exercise with a series of unsurprising findings and general recommendations that could have been gleaned from the literature. The ms discussion ends with "...while CFIR constructs should ideally be coupled to specific targeted outcomes, the formative nature of this study precludes linkage of implementation determinants to outcomes". Possibly this is the crux of my concern and perhaps authors could elaborate. After all, authors stated at the outset that the study was conducted to evaluate the implementation climate for cMDA and to determine barriers and facilitators to launch".

In implementation research, we often want to pair implementation determinants (facilitators and barriers) with implementation outcomes, to compare the relative impact of a determinant on implementation (in this case, cMDA coverage). However, because this is a formative study, and not a process or summative evaluation, we did not pair determinants with implementation outcomes. We have amended this sentence to make this more clear, and it now reads as, "Additionally, because we conducted a formative study we did not link identified implementation determinants to observed implementation outcomes, however subsequent data collection activities in DeWorm3—once the trial is underway and outcome data are collected—will afford these opportunities."

REVIEWER 3: Dr. Nor Asiah Muhamad, Institute for Public Health

Comments to the Author:

1. The description of your methodology part is confusing, the conceptual flaws in my opinion need to be improved.

Thank you for your feedback. The methods section presented aligns with best practices in the Consolidated Criteria for Reporting Qualitative Research (COREQ) checklist for publishing qualitative studies. We have made several updates to the methods section to try to ensure clarity and transparency. If there are other specific aspects of the methodology that the reviewer finds particularly confusing and would like us to update, we would be happy to address those specific components.

2. As you are mentioning the formative qualitative research can be used to understand community-member and implementer perceptions in community-based campaigns and diagnostic analyses, it remains unclear how these processes works and can facilitate the application of formative evaluations. Maybe you can explain further in the method.

Thank you for this feedback. Formative research, and diagnostic analysis in particular with are an application of formative research, help researchers preemptively understand potential facilitators and barriers that will affect rollout of a new intervention or public health program. They can help identify bottlenecks in implementation, ideally before they occur. This helps save time, resources, and avoids implementation pitfalls that could undermine community trust in an intervention.

We have added the following italicized text to the introduction of the paper to ensure clarity, "Formative qualitative research can be used to understand community-member and implementer perceptions of past, ongoing, or prospective community-based campaigns.

Diagnostic analyses, an application of formative evaluations, are particularly helpful in illuminating processes that can facilitate or impede implementation. *Diagnostic analyses help to identify determinants of current practices, potential barriers and facilitators to implementing new interventions, and the perceived feasibility or utility of a new implementation strategy. This formative evidence can help researchers and implementers understand potential implementation challenges and, ideally, address them prior to intervention launch.* In this study, we perform a diagnostic analysis of the implementation climate to proactively identify factors influencing the launch of cMDA for STH transmission interruption, including (1) perceptions of current deworming practice, (2) potential barriers and facilitators to transitioning from school-based MDA to cMDA delivery, and (3) perceived effectiveness and need for cMDA (13).”

3. The research gaps are not clearly described. What makes the study contribution and innovation unclear: is the gap seen in the methodological design of merely cross-sectional research? Or, in the used concepts and model? Please mention the gaps.

We have added more detail about limitations to the discussion section. While generally the cross-sectional nature of qualitative data collection is not considered a limitation, potential gaps include social desirability biases and potential challenges in generalizability. We have added the following text to the paper: “This study had several limitations. Some participants may have heard about DeWorm3 before participating in FGDs, which may have contributed to social desirability or response biases. While a large number of FGDs were conducted across heterogeneous settings, it is also possible that the study findings may not be generalizable to other STH-endemic areas.”

4. How is randomisation done? What method is used?

Random participant selection was conducted for community members in Benin and Malawi. We have expanded upon this section of the methods with the following text, “In Malawi, community members were randomly selected to participate from a pool of individuals who attended outreach meetings at the chiefs/headmen’s residence. The first five randomly approached individuals from each demographic strata who agreed to participate were invited to attend FGDs within the next week (except children, for whom parents/caregivers were approached). In Benin, community members were selected from a randomly generated list of potential participants from a baseline census database. The research team contacted the household heads by telephone and invited a specific individual (woman, man, or child) to participate in an FGD. No more than one individual per household was selected to participate in an FGD in a given cluster.”

VERSION 2 – REVIEW

REVIEWER	Nor Asiah Muhamad Institute for Public Health
REVIEW RETURNED	17-May-2022
GENERAL COMMENTS	In my opinion this work is ready for publication